# Effect of Sulodexide on Circulating Blood Cells in Patients with Mild COVID-19

**DOI:** 10.3390/jcm11071995

**Published:** 2022-04-02

**Authors:** Arthur Melkumyants, Lyudmila Buryachkovskaya, Nikita Lomakin, Olga Antonova, Julia Docenko, Vladimir Ermishkin, Victor Serebruany

**Affiliations:** 1National Medical Research Center of Cardiology, Moscow 121552, Russia; livbur@mail.ru (L.B.); loa_lu@mail.ru (O.A.); docenko_ulia@mail.ru (J.D.); v.v.erm@mail.ru (V.E.); 2Institute of Physics and Technology, Moscow 141701, Russia; 3Cardiology Division, Central Clinical Hospital of Presidential Administration, Moscow 121359, Russia; nikitalomakin@gmail.com; 4Division of Neurology, Johns Hopkins University, Baltimore, MD 14110, USA; vserebr1@jhmi.edu or

**Keywords:** sulodexide, endothelial cells, platelet activation, erythrocyte sludges, COVID-19, inflammation

## Abstract

Background. Despite the fact that COVID-19 usually manifests with severe pneumonia, there is a growing body of evidence that life-threatening multiorgan damage is caused by vascular and hemostatic abnormalities. Since there is no established therapy, assessing antithrombotics is indeed important. Sulodexide, a compound derived from porcine *intestinal mucosa* is a mixture of fast-moving heparin fraction (80%) and dermatan sulfate (20%), is approved in Europe and currently in trials for COVID-19 indication. Methods. This single-center, prospective, observational study included 28 patients with mild COVID-19 hospitalized in the Central Clinical Hospital of the Presidential Administration of the Russian Federation. Patients in the control group (*n* = 14) were treated using routine therapy according to current guidelines, while patients in the experimental group (*n* = 14) had the routine treatment supplemented with daily intravenous injections of sulodexide in 600-unit doses. Scanning electron microscopy was utilized to examine the blood specimens derived from the cubital vein at admission and at 10 days after hospitalization, which was approximately the average duration of patients’ treatment in the hospital (11.6 ± 0.4 days). Results. Sulodexide significantly (by 40%) diminished the score of circulating endothelial cells, potentially indicating its antiviral endothelium-protective properties. It also prevented the extra activation of the platelets and the formation of erythrocytic sludges. Among patients in the control group, the share of activated platelets rose from 37 ± 5% to 45 ± 6% (*p* = 0.04) over the course of the study period, whereas among patients in the experimental group, the share of activated platelets remained practically unchanged (43 ± 6% vs. 38 ± 4%, *p* = 0.22). The score of erythrocytic sludges in the control group remained practically the same (4.8 ± 1.1 at admission vs. 3.9 ± 0.9 after 10 days, *p* = 0.67), whereas in the experimental group, it significantly decreased (from 5.7 ± 1.7 to 2.4 ± 0.9, *p* = 0.03). Conclusions. Sulodexide is able to defend endothelium, normalize blood, and, seemingly, prevent thrombosis. Therefore, it may be considered as a promising and effective agent for the treatment of patients with mild COVID-19. Broader randomized trials are needed to assess whether the observed findings will transform into sustained long-term clinical benefit.

## 1. Introduction

Since the end of 2019, the world has been plagued with the novel coronavirus disease COVID-19 (the pandemic was announced by the WHO on 11 March 2020), which has had millions of victims and continues to grow. In fact, more than 5 million people had died of COVID-19 by the end of fall in 2021 alone. This previously unknown disease attracted special attention since in 10–15% patients it provoked severe pneumonia accompanied with a high-grade fever, severe dry cough, and dyspnea [1]. 

Unfortunately, no pathogenetic remedies are currently available to cope with this infection, so it continues to create its huge amount of sorrow. In parallel with lung damage, pronounced thrombosis occurs in microcirculation. In fact, accumulating clinical experience suggests that COVID-19 death is characterized by not only lesions to the lungs but also severe multiorgan failures [2]. 

By damaging the vascular endothelium and provoking inflammation, the virus causes the development of a “cytokine storm”, which in its turn contributes to vascular endothelial lesions and augments the inflammatory reaction [3]. These endothelial lesions lead to coagulopathy [4] and the development of systemic vasculitis and thrombosis, which disturb the normal blood supply to the organs and finally result in the development of multi-organ failure [5,6]. It is generally accepted that these pathologies are underlain by the progression of coagulopathy and endotheliitis. Indeed, endothelial damage is the key factor associated with the devastating effect of COVID-19 on the functions of various organs. Consequently, an effective treatment strategy for COVID-19 should not only involve directly annihilating the virus or suppressing its replication, but also protect the vascular endothelium.

Sulodexide [7] is a promising endothelial protector that is produced by Alfasigma in Italy and branded as Vessel Due F, currently approved in Europe. Sulodexide is a compound derived from porcine small intestine mucosa. It is a natural mixture of the fast-moving heparin-like fraction (80%) and dermatan sulfate (20%). Sulodexide has been shown to protect endothelial glycocalyx (GC), which is a protective layer of proteoglycans covering the luminal surface of endothelial cells (EC). Sulodexide has also been shown to restore endothelium integrity after lesions [8]. Numerous studies have demonstrated that sulodexide not only is an anticoagulant but also has anti-aggregating, antithrombotic, fibrinolytic, and angioprotective effects [9,10].

With such remarkable properties, sulodexide may prevent the development of endothelial lesions in patients with COVID-19 and hence may inhibit the progression of inflammation and coagulopathy. To test this hypothesis, we conducted an electron microscopic study of the blood specimens drawn from COVID-19 patients treated with sulodexide as a supplement to routine therapy.

## 2. Methods

The prospective observational single-center study included 28 patients with polymerase-chain-reaction-confirmed COVID-19 diagnosis who were treated in the Central Clinical Hospital of the Presidential Administration (Moscow) from 25 May to 22 July 2020. Patients of moderate severity with a chest CT score of 1 or 2 were consistently included in the study. Patients who required an ICU admission were excluded from the study. The included patients were divided into two groups. In the control group (*n* = 14), patients were treated using routine therapy according to current guidelines, while in the experimental group (*n* = 14), the routine treatment was supplemented with daily intravenous injections (2 mL) of sulodexide (Alfasigma, Italy) in the dose of 600 units. 

Baseline body temperature, ECG, heart rate (HR), arterial pressure (AP), respiration rate (RR), and peripheral oxygen saturation (SpO_2_) were obtained on admission. In venous blood, the following biochemical indices were determined: pH, hepatic enzymes, creatinine, and lactate. Inflammation was assessed using C-reactive protein (CRP), erythrocyte sedimentation rate (ESR), fibrinogen, interleukin-6, ferritin, and procalcitonin. Coagulation was assessed using D-dimer, activated partial thromboplastin time (aPTT), prothrombin time (PT), and anti–factor Xa.

The patients were treated with antiviral drugs favipiravir or hydroxychloroquine. The pro-inflammatory cytokines were inhibited with tocilizumab [11], olokizumab [12], and levilimab. Oxygen therapy was also used. To prevent thrombosis, all patients were subcutaneously injected with enoxaparin (two patients with preventive dose (40 mg/day), and 26 patients with intermediate (60 mg/day) or treatment (80 mg/day) doses).

A 0.5 mL blood sample was collected from each patient for scanning electron microscopy on admission as well as after 10 days of treatment. During the therapeutic course, the abovementioned respiratory and cardiovascular parameters were recorded daily. The blood was drawn for clinical analyses at least once every three days. At the end of the study, we compared the clinical analyses and electron microscopic data of the blood specimens taken on admission and 10 days later in both the control and sulodexide groups.

All patients gave informed consent for the research study, which was carried out in strict adherence to ethical directives and regulations of World Medical Association Declaration of Helsinki. The research protocols were approved by Ethical Committee of the Central Clinical Hospital of the Presidential Administration.

**Blood specimens for electron microscopy**. Blood was drawn from the cubital vein into VACUETTE tubes (Greiner bio-one, Austria) containing sodium citrate (3.2%) with an anticoagulant:blood ratio of 1:9. The fixed blood specimens (20 µL) were then sedimented on Nucleopore polycarbonate membrane filter (d = 25 mm, pore size 0.2–0.4 µM, Nucleopore Company Corporation, Pleasanton, CA, USA). Additional details of this method have been previously described elsewhere [13]. A quarter of the filter was cut out, and it contained the cells sedimented from a 5 µL blood specimen. Through consecutive scanning of the entire surface of this quarter of the filter, we counted the total number of circulating endothelial cells (CEC), which then was recalculated for 1 mL blood. All the blood cells (erythrocytes and their sludges, platelets, leucocytes) and their aggregates were investigated in the same specimens. All these cells were counted on 25 squares sizing 10 × 10 µm and placed diagonally. The data are summarized as the mean obtained in 10 scan fields. The blood specimens were examined under an Inspect F50 FEI scanning electron microscope with X-Max EDS-Detector (Oxford Instruments).

**Statistics**. The continuous data are represented as m ± SEM. Categorical variables are described as percentages and were compared by Chi-square test. Since the differences between the data obtained in sulodexide-treated patients and in the control ones were especially interesting, the values recorded prior to the first therapy day and after 10 days of treatment were compared statistically using the Mann–Whitney non-parametric test. All tests were two-tailed, and *p* < 0.05 was considered statistically significant. Statistical data are represented in the form of a comparison between values obtained on admission and after 10 days of treatment for control and sulodexide groups.

## 3. Results

There were no ICU admissions among the patients included in the study during the course of the study period. Patients treated with sulodexide in addition to routine care did not demonstrate any symptoms of bleeding. 

The patients included in the study were hospitalized 5.9 ± 0.8 (1–11) days after manifestation of the first symptoms of the disease and stay in the hospital during 11.6 ± 0.4 days. On admission, the severity of illness was moderate according to pulse oximetry and pulmonary CT. The admission parameters of the patients are summarized in Table 1.

**Basic biochemical and physiological parameters.** The basic clinical parameters determined on admission and discharge from the hospital are summarized in Table 2. 

As can be seen from the data given in the table, sulodexide in our study had an effect only on neutrophils and C-reactive protein; the latter decreased in patients receiving this drug significantly faster than in patients in the control group. Also noteworthy is the effect of sulodexide on D-dimer level. It is possible that the patient sample was too small to demonstrate a significant difference in D-dimer level between the two groups, since a trend toward a decrease in the D-dimer level in the sulodexide group compared to the control group was observed. The change in all other biochemical parameters in both groups is almost the same. (A comparison of changes in all parameters except CRP and D-dimer gives a *p* value of at least 0.3.) In contrast, the data obtained by electron microscopy revealed significant differences between the patients in the control and experimental groups. 

***1. Circulated******endothelial cells (CEC).*** It is known that in healthy individuals, the level of CEC does not exceed three cells per milliliter [14]. Prior to therapy, the CEC level in the group treated with sulodexide was 2.2 ± 0.3 cells per quarter of polycarbonate membrane filter (ranging 1–7 cells) and 2.0 ± 0.3 cells (1–6 cells) per quarter in the control group. Since this part of filter was covered with 5 µL blood, a simple calculation gave the content of CEC per 1 mL blood. Thus, the CEC level in the experimental group prior to the therapy was 440 ± 60 cell/mL. After 10 days of therapy with sulodexide, the CEC level significantly (*p* = 0.006) decreased to 1.3 ± 0.3 cells (ranging 0–3 cells) per quarter, which corresponded to 270 ± 60 cell/mL.

In contrast, the quantity of CEC in the control group observed on a quarter of polycarbonate membrane filter increased after the therapy from 2.0 ± 0.3 (1–6 cells) to 2.4 ± 0.3 (1–6 cells), corresponding to an increasing trend in CEC level from 400 ± 60 to 470 ± 70 cell/mL (Figure 1A); however, this increase did not reach statistical significance (*p* = 0.28).

On admission, the membranes of all CEC demonstrated numerous perforations with the diameter of SARS-CoV-2 capsid. Treatment with sulodexide significantly decreased the number of such perforations down to their complete disappearance. Similar findings were observed only in 3 of 14 patients (21.4%) in the control group. The effect of sulodexide on CEC levels in the blood and the morphological status of CEC estimated by scanning electron microscopy are shown in Figure 1A,B.

***2. Erythrocytic aggregates (sludges).*** On admission and after 10 days of treatment, the blood of control and experimental patients contained clearly visible, long erythrocytic coin-roll formations (sludges) composed of packed erythrocytes (Figure 2A,B), which is not characteristic of healthy subjects. Prior to the initiation of the therapy, the number of sludges scored within a square of 10 × 10 µm in experimental and control groups were 5.7 ± 1.7 and 4.8 ± 1.1, respectively (*p* > 0.3). After 10 days of treatment in control patients, the number of sludges in the square insignificantly decreased to 3.9 ± 0.9 (*p* = 0.67), corresponding to a similarly insignificant drop in the number of cells forming a sludge. In contrast, the sulodexide group demonstrated a significant decrease in the quantity of sludges per square down to 2.4 ± 0.9 (*p* = 0.03), and in the number of erythrocytes per sludge down to 2–3 cells (Figure 2A,B).

***3. Activated platelets*.** The activated platelets are characterized with a spherical form with pseudopodia, and these features make it easy to differentiate them in microscopic studies from normal non-activated cells. On admission in seven patients (25% of all patients), concentration of activated platelets was normal (less than 5%), whereas 10 days later, a normal concentration of activated platelets was found in six patients (21.4%). However, in the experimental group treated with sulodexide, the share of activated platelets practically did not change after a 10-day treatment, being 43 ± 6% on admission and 38 ± 4% after termination of sulodexide therapy (*p* = 0.22). In contrast, the routine 10-day-long therapy in the control group was characterized by a significant increase in the share of activated platelets from 37 ± 5% to 45 ± 6% (*p* = 0.04). The changes in relative content of activated platelets in the blood of patients with COVID-19 are shown in Figure 3A,B.

***4.******Platelet aggregates***. An increased number of activated platelets in the control group was not associated with an elevated count of platelet–erythrocyte aggregates (PEA) among these patients. While in the experimental group, this parameter practically did not change over 10 days of treatment (29 ± 4 vs. initial value of 23 ± 4; *p* = 0.36), the count of PEA in the control group significantly increased from 23 ± 4 to 38 ± 4 (*p* = 0.04). During 10 days of therapy, there was no significant change in the quantity of leukocyte-platelet aggregates and echinocytes in either group (*p* = 0.48 and *p* = 0.34, respectively). 

## 4. Discussion

The primary aim of this study was to demonstrate that sulodexide may be an effective remedy for COVID-19. Despite the fact that treatment with sulodexide does not improve the clinical state of patients as compared with routine treatment, our data demonstrate a protective effect of sulodexide on the state of endothelial cells, which may have a beneficial hematologic effect.

As we mentioned earlier, initial clinical attention to COVID-19 focused mostly on the pulmonary system, because severe pneumonia was the prevailing manifestation. However, after it had been established that this infection could affect virtually all organ systems, it became evident that the virus invades the organism through the vascular network and that the endothelium plays the most important role in organ damage [5]. 

Starting from the classical work be Furchgott and Zawadszki [15], the endothelium has been viewed as playing an increasingly important role in the effects of the vascular system. Specifically, it was established that this monolayer of cells covering all surfaces that have contact with blood plays crucial roles not only in the control of vascular tone and homeostasis but also in assisting immune reactions. The connection between the endothelium and immunity is based on the observations that (1) endothelial cells are the targets for most viruses and (2) endothelial lesions associated with EC desquamation are typical of numerous critical states such as severe sepsis and viral infections [5,16], which trigger immune reactions culminating in a cytokine storm [6]. These pathologies are provoked by COVID-19.

The leading role of endothelium in the spread of novel coronavirus disease is explained by the fact that ACE2 receptors, which are the targets of SARS-CoV-2, are expressed on endothelial cells of blood vessels in almost all organs. This reasoning suggests a mechanism for the spread of this infection in humans. After entering the alveoli through airways, the coronavirus damages them and disrupts the integrity of the alveolar–capillary barrier. The virus subsequently invades the pulmonary circulation and travels with the blood to all organs, where it binds with and penetrates the endothelial cells. After replication in endothelial cells, SARS-CoV-2 kills the host cell. Thereafter, the damaged endothelial cells detach from the vascular wall, thereby denuding the thrombogenic and proinflammatory subendothelial surface, which in its turn results in the development of coagulopathy, perivascular inflammation, tissue edema, and a procoagulant state.

Importantly, in order to attach to endothelial cells and invade it, the virus must overpass the protective endothelial layer, i.e., the GC. This macromolecular layer, composed of proteoglycans, glycosaminoglycans, glycoproteins, and glycolipids, covers the luminal face of endothelial cells [17,18], and it is the GC in combination with the endothelium that provides homeostasis of vascular networks by regulating the vascular permeability and tone, preventing the development of microvascular thrombosis, and suppressing the adhesion of leukocytes with the platelets. Evidently, the destruction of GC should lead to endothelial dysfunction [19,20], to the development of edema due to impaired capillary permeability, to the inflammation of the vascular wall, and finally to the hypercoagulation and paresis of vascular tone regulation [21].

It is common knowledge that endothelial GC is degraded in some pathological states. In particular, GC lesions are produced by viruses and especially by sepsis [22,23]. It is also known that endothelial GC lesions are a risk factor to the development of COVID-19 [24,25]. Based on these data, it seems logical to counteract the devastating effects of SARS-CoV-2 with the search for a way to prevent damage to GC and endothelial cells, thereby minimizing disturbances in homeostasis.

Sulodexide appears to be the most promising way to cope with endothelial lesions protecting GC and, consequently, endothelium [26]. Studies carried out on cell cultures, animals, and humans [27,28,29] showed that sulodexide strongly counteracts the inflammatory process by suppressing the release of proinflammatory cytokines and chemokines [28]. In addition, by ameliorating the oxidative stress via the down-regulation of ROS production and up-regulation of superoxide dismutase synthesis, sulodexide counterbalances the damage to GC. In addition, the composition of sulodexide explains its antithrombotic effect, which suppresses platelet activation in response to the action of thrombin and tissue factors, thereby exerting an effect similar to that of enoxaparin [30,31].

Finally, a very important factor is the safety of antithrombotic agents such as sulodexide [32,33]. Although sulodexide in this study was administered intravenously without any adjustments to the dose of low-molecular-weight heparin injected subcutaneously, we did not observe even minor (gingival or nasal) bleeding.

The data on anti-inflammatory and blood-normalizing effects of sulodexide were corroborated by our study. In fact, the quantity of CEC in the sulodexide group decreased by 40% after 10 days of treatment, in contrast to control group, in which no such effect was observed (Figure 1). Taking into consideration that the quantity of CEC can serve as a marker and a quantitative measure of endothelial damage, the present data demonstrated that sulodexide could protect endothelial cells against viral aggression. By decreasing the degree of vascular denudation, sulodexide should ipso facto diminish the area of the denuded surface of the vessels, which consequently should ameliorate inflammation. In turn, the inhibition of vascular inflammation should limit the disturbances in the structure of the blood (Figure 2). In fact, after a 10-day treatment, the score of erythrocytic sludges in experimental group dropped almost three-fold, while in the control group, it remained virtually unchanged. Remembering that erythrocytic coin-roll formations impede normal blood supply because they cannot travel through the capillaries and secure adequate gas exchange in the tissues, one can conclude that sulodexide can normalize or at least pronouncedly improve blood supply to the organs, thereby diminishing the degree of tissue hypoxia.

Importantly, sulodexide suppressed the activation of the platelets and diminished the degree of hypercoagulation, while in the control group, the score of activated platelets increased persistently despite uninterrupted routine treatment (Figure 3). It should be stressed that we started to use sulodexide in the treatment of patients with COVID-19 rather late. The reason was that these patients arrived to the hospital as late as 5.9 ± 0.8 days after manifestation of the first symptoms of the disease. This is a sufficiently long time for the virus to provoke marked inflammation in the vascular wall and coagulopathy. It is possible that, if we used sulodexide earlier, its effect could have been more pronounced. 

This hypothesis was corroborated by Mexican researchers, who started the sulodexide-based treatment of COVID-19 patients in the outpatient setting no later than 3 days after the appearance of symptoms [34]. They treated 124 unvaccinated patients (experimental group) for 21 days with daily oral sulodexide (1000 U), while 119 patients (the control group) were “treated” with placebo. In this study, it was shown that treatment with sulodexide significantly reduces the need for hospitalization and for respiratory support. These results attest to the effectiveness of sulodexide in the treatment of COVID-19 patients in outpatient clinics at the early stage of the disease.

## 5. Limitations 

Evidently, the numbers of patients in this study and another publication [34] are far from being sufficient to prove comprehensively the therapeutic effect of sulodexide in the treatment of patients with COVID-19. The small patient sample size also limited qualitative statistical analyses of the results. Nevertheless, the data in both studies attest to the usefulness and even the necessity of carrying out a large-scale multicenter examination of sulodexide effects in patients with vascular damage caused by SARS-CoV-2. 

## 6. Conclusions

Sulodexide, which possesses numerous beneficial pharmacological features such as a good safety profile and the possibility of oral administration, can probably cope effectively with vascular inflammation and significantly diminish disturbances in the composition and function of the blood in COVID-19 patients. Our data suggest that sulodexide may be a promising agent, which can serve for the safe and effective treatment of patients infected with SARS-CoV-2. However, it is obvious that broader randomized trials are needed to assess whether the observed findings will transform into sustained a long-term clinical benefit.

## Figures and Tables

**Figure 1 jcm-11-01995-f001:**
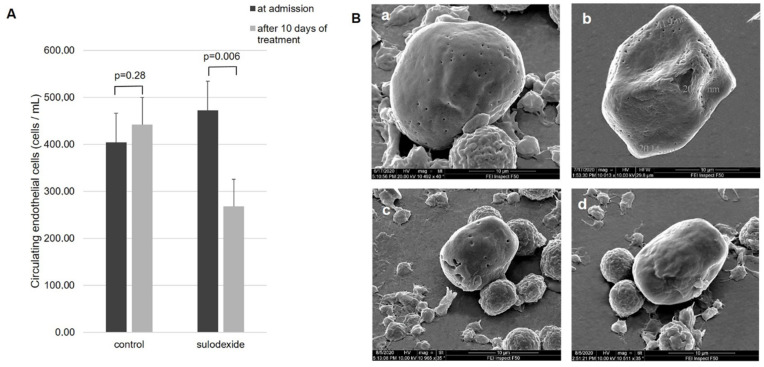
(**A**): Effect of sulodexide on CEC level. (**B**): CEC in the blood of patients on admission with membrane fenestration typical of COVID-19 (**a**,**c**). After routine treatment of control patients, the quantity of such CEC did not decrease. Moreover, their morphological features (characterized by a large number of membrane fenestrations with the size of the viral capsid) were retained (**b**). The addition of sulodexide to the standard treatment significantly decreased the quantity of CEC in blood specimens and decreased the number of virus-produced fenestrations to the extent of their complete disappearance (**d**). Magnification ×10.000.

**Figure 2 jcm-11-01995-f002:**
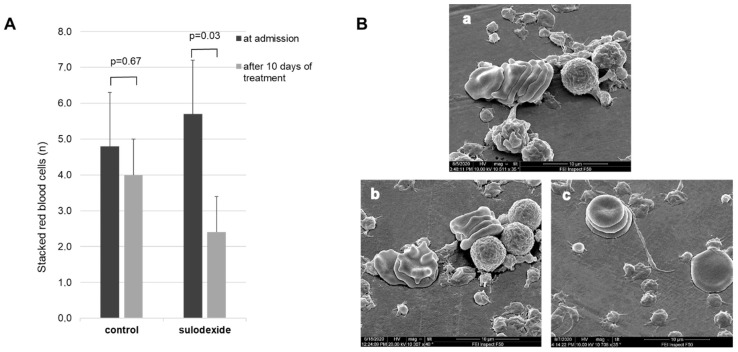
(**A**): Effect of routine treatment and treatment including sulodexide on the quantity of erythrocytic sludges in venous blood of patients with COVID-19. (**B**): Microphotographs of blood specimens on admission (**a**) and after 10 days of treatment in control (**b**) and experimental (**c**) groups. Magnification ×10,000.

**Figure 3 jcm-11-01995-f003:**
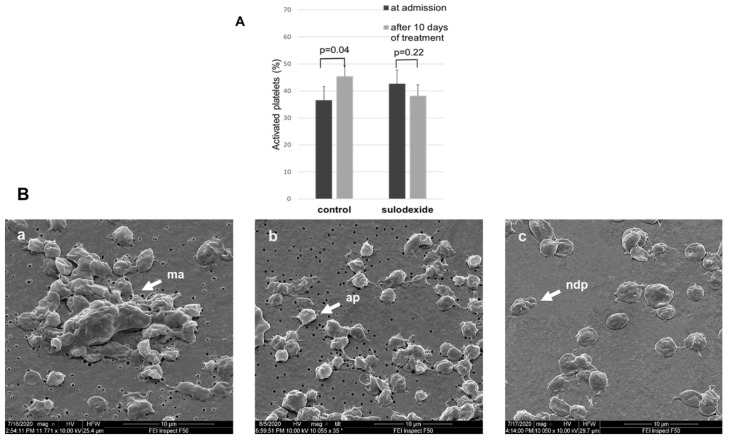
(**A**): Effect of routine treatment and treatment including sulodexide on the share of activated platelets in venous blood of patients with COVID-19. (**B**): Microphotographs of blood specimens on admission (**a**) and after 10 days of treatment in control (**b**) and experimental (**c**) groups. Arrows indicate ma—microaggregates; ap—activated platelets; ndp—normal discoid platelets. Magnification ×10,000.

**Table 1 jcm-11-01995-t001:** Admission parameters of the patients.

Parameter	Controls (*n* = 14)	Sulodexide (*n* = 14)	*p*-Value for Factor Homogeneity Cross Samples
men/women, *n* (%)	7 (50%)/7 (50%)	7 (50%)/7 (50%)	-
Age (years)	47.4 ± 3.5 (32–71)	59.2 ± 2.7 (35–74)	0.02
BMI (kg/m^2^)	25.3 ± 2.4 (21.8–31.4)	24.7 ± 2.9 (22.4–32.9)	0.28
Length of stay in hospital (days)	11.7 ± 1.8 (9–20)	13.8 ± 1.2 (10–19)	0.18
Obesity	5 (35.7%)	6 (42.8%)	0.49
Smokers	4 (28.6%)	0	0.04
Hypertension	4 (28.6%)	10 (50%)	0.02
Diabetes	0	2 (14.3%)	0.12
Coronary artery disease	1 (7.1%)	3 (21.4%)	0.05
Heart failure	1 (7.1%)	1 (7.1%)	0.56
Cancer	2 (14.3%)	6 (42.8%)	0.02
Chronic kidney disease	0	1 (7.1%)	0.33
COPD	0	0	–

**Table 2 jcm-11-01995-t002:** Basic biochemical and physiological parameters of COVID-19 patients before and after treatment.

Variable	Controls (*n* = 14)		*p*	Sulodexide (*n* = 14)		*p*	Experiment/Control Difference on Discharge
	On Admission	After 10 Days of Treatment		On Admission	After 10 Days of Treatment		
Heart rate (min^−1^)	87.5 ± 3.5	72.2 ± 3.8	0.01	89.2 ± 3.7	78.3 ± 2.9	0.02	ns
Respiratory rate (min^−1^)	17.0 ± 0.7	16.4 ± 0.3	0.07	18.8 ± 0.4	17.4 ± 0.4	0.04	ns
Temperature (°C)	37.5 ± 0.2	36.5 ± 0.2	0.001	37.4 ± 0.2	36.4 ± 0.1	0.002	ns
SpO_2_ (%)	97.8 ± 0.3	98.3 ± 0.1	0.53	96.7 ± 0.3	97.8 ± 0.4	0.44	ns
Hemoglobin (g/L)	138.6 ± 5.2	136.7 ± 7.3	0.19	135.9 ± 5.8	131.0 ± 7.8	0.11	ns
Erythrocytes (10^12^/L)	4.1 ± 0.6	4.3 ± 0.5	0.38	4.4 ± 0.6	4.4 ± 0.4	0.58	ns
Leukocytes (10^9^/L)	5.7 ± 0.6	5.9 ± 0.9	0.34	6.4 ± 0.7	6.2 ± 0.6	0.87	ns
Platelets (10^9^/L)	212.9 ± 10.7	251.0 ± 27.2	0.04	196.6 ±16.6	244.6 ± 27.3	0.05	ns
Lymphocytes (%)	25.6 ± 2.6	33.5 ± 2.8	0.01	24.2 ± 2.5	27.4 ± 1.8	0.07	ns
Neutrophils (%)	61.1 ± 2.7	48.1 ± 3.4	0.001	66.4 ± 2.9	60.6 ± 3.9	0.12	0.03
ESR (mm/h)	26.8 ± 7.8	26.5 ± 4.9	0.68	34.9 ± 8.2	32.3 ± 6.3	0.72	ns
CRP (mg/L)	17.7 ± 7.6	4.4 ± 2.4	0.05	19.1 ± 2.7	1.4 ± 1.2	0.03	0.04
Creatinine (mg/dL)	90.0 ± 2.0	87.4 ± 2.9	0.28	92.6 ± 6.1	101.6 ± 12.7	0.12	ns
D-dimer (ng/L)	291.2 ± 38.7	219.4 ± 42.8	0.08	313.4 ± 42.3	168.9 ± 37.4	0.05	0.06
Fibrinogen (g/L)	4.9 ± 0.3	4.0 ± 0.5	0.14	4.4 ± 0.2	4.2 ± 0.3	0.59	ns
Ferritin (µg/L)	244.9 ± 57.4	327.8 ± 59.7	0.11	270.7 ± 54.7	409.1 ± 45.3	0.08	ns

## Data Availability

The data presented in this study are original.

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
