# Peer review of "Effect of Sulodexide on Circulating Blood Cells in Patients with Mild COVID-19"

_jcm, 2022, doi:10.3390/jcm11071995_

Round 1
Reviewer 1 Report
The present manuscript from Melkumyants A. et al reports data showing the effect of sulodexide on circulating blood cells in patients with COVID-19. This is an interesting article, but there are some points that need to be discussed.
- Could the authors provide the inclusion and exclusion criteria for the recruitment of the patients?
- Could the author strengthen the observations done on circulating blood with scanning electron microscopy before and after sulodexide treatment with other evaluations (e.g. marker of endothelial damage, platelet activation..)?
- In material and methods section, could the authors add which test was used to confirm SARS—COV-2 infection? It was confirmed by a reverse-transcriptase polymerase chain reaction?
4. Could the authors add how they prepare samples to evaluate erythrocytic aggregates and platelet aggregates? If I understand correctly is the same procedure for the evaluation of circulating endothelial cells, but this point needs to be clarified in material and methods section
Author Response
Please, see the attachment

Reviewer 2 Report
The paper of Melkumyants et al. is dealing with a hot-topic domain, namely treatment of Covid-19 patients. The authors aim to study effects of an antithrombotic compound sulodexide on blood cells as well as on desquamation of the endothelial cells in patients having moderate Covid-19. The results obtained show that sulodexide treatment in the period 1-2 weeks after manifestation of symptoms has beneficial effects on the endothelial cells and circulating cells. There data could be interesting for readers searching for efficient therapeutic approaches to treat the disease.
General remark.
My main concern here is that the authors do not determine the parameters of interest (activated platelets, endothelial cell desquamation, red blood cell sludges etc.) in control healthy subjects. Therefore, the reader cannot compare different values under conditions of disease/treatment with normal values obtained by the same operators. I can understand that the authors tried to get their results as quickly as possible keeping in mind that the topic is very hot; anyway, such a disadvantage decreases the general value of study.
Special remarks.
There is a general confusion in the paper - the authors use the term « routine treatment” for 2 different treatments: 1) treatment got by all patients, and 2) placebo treatment. See e.g. the legend for the Fig.3 “Effect of routine and experimental treatment…” This is not correct, all patients had routine treatment.
The Introduction section could be condensed.
Method section.
“Coagulation was assessed with D-dimer, aPTT, PT, and anti-Xa.” Please explain the abbreviations. The reviewer did not find values of PTT and anti-Xa in the text.
A very important point is the statistics, because the paper uses 2 different types of comparison – between sulodexide and placebo as well as between starting point and end of the treatment. Statistical methods should be specified in detail. For example, the tiny difference in respiratory rate (Table 2) between admission time (17.0 ± 0.7) and 10 days later (16.4 ± 0.3) reaches p value as low as 0.01. This should be explained (repeated measures?);
Methods of statistical comparison between sulodexide and placebo groups should be specified separately.
Results.
p.5, first para. “The treatment with sulodexide significantly diminished the number of such perforations down to complete disappearance of them, which was observed only in 3 of 14 patients in the control group.” The decrease in number of patients having cell perforations should be estimated statistically.
p.5, last para. Number of red blood cells forming a sludge could be an interesting parameter. The authors should quantify it and show the statistics.
Fig. 3B. Please indicate typical activated and non-activated platelets by arrows.
The last sentence of the Result section. “During a 10-days-long therapy, the scores of leukocyte-platelet aggregates and echinocytes in both groups changed insignificantly (p>0.3).” Please provide values for this parameter.
Discussion.
The first sentence of the Discussion section. “The main finding of this study is to demonstrate that sulodexide may be an effective remedy during curing of COVID-19.” This statement is too strong. The authors provide no data showing an improvement of the clinical state of patients by the drug and could only conclude that sulodexide has beneficial effects on some blood properties.
Similarly, the last sentence of the second para “endothelium plays the most important role in the damage to the organs” is not completely true. The endothelium is only a gate for the virus invasion in Covid-19, it is not correct to say that its role is primordial, because many other types of cells are also involved in the pathogenesis.
Minor remarks.
Please indicate in the Abstract the treatment duration.
Do the authors use placebo injections for the control group?
Please add numbers into the description of activation of platelets and red blood cell sludges to make the results more quantitative in the abstract.
p.4, second para from the bottom, 3rd line. Why “in contrast to”? The values for both groups are similar ones.
Figure 1A was referenced two times in the text
In the Discussion section, the authors describe in many details (including statistics, number of patients etc.) a paper of Mexican researchers. In my opinion, there is no need to devote a relatively large paragraph to an extremely detailed description of another study. Better to speculate more on possible cell mechanisms of sulodexide effects.
Please improve English and correct some typos.
Reviewer 3 Report
Interesting study looking at adjunctive treatment to decrease endovascular damage, given in combination with VTE prophylaxis and antivirals (although hydroxychloroquine no longer recommended). Would also include whether pts were treated with steroids since this is often a part of their treatment.
Study evaluates utility of sulodexide by measuring difference in circulating endothelial cells, erythrocyte aggregates, activated platelets and platelet aggregates, however there is no background explaining these tools in evaluating endothelial damage. Would include rationale for using these as well as any evidence supporting.
Would also include incidence of thrombosis (arterial and venous) in both groups.
Round 2
Reviewer 1 Report
I thank the authors for the answers to my comments. I am satisfied with their reply.